# Innovations in Antifungal Drug Discovery among Cell Envelope Synthesis Enzymes through Structural Insights

**DOI:** 10.3390/jof10030171

**Published:** 2024-02-22

**Authors:** Yue Zhou, Todd B. Reynolds

**Affiliations:** Department of Microbiology, University of Tennessee, Knoxville, TN 37996, USA; yzhou58@vols.utk.edu

**Keywords:** cryo-EM, structure biology, membrane-bound enzymes, antifungal development, drug resistance, rational drug design

## Abstract

Life-threatening systemic fungal infections occur in immunocompromised patients at an alarming rate. Current antifungal therapies face challenges like drug resistance and patient toxicity, emphasizing the need for new treatments. Membrane-bound enzymes account for a large proportion of current and potential antifungal targets, especially ones that contribute to cell wall and cell membrane biosynthesis. Moreover, structural biology has led to a better understanding of the mechanisms by which these enzymes synthesize their products, as well as the mechanism of action for some antifungals. This review summarizes the structures of several current and potential membrane-bound antifungal targets involved in cell wall and cell membrane biosynthesis and their interactions with known inhibitors or drugs. The proposed mechanisms of action for some molecules, gleaned from detailed inhibitor–protein studeis, are also described, which aids in further rational drug design. Furthermore, some potential membrane-bound antifungal targets with known inhibitors that lack solved structures are discussed, as these might be good enzymes for future structure interrogation.

## 1. Introduction

Over the past three decades, fungal infections in humans have surged due to a rise in immunocompromised patients [1,2,3]. The most common culprits behind these infections are *Candida* species, with *Candida albicans* being the most prevalent. *C. albicans* is known for causing a wide range of infections, including those affecting the mucous membranes, skin, and bloodstream [2,4,5]. Effective treatment requires antifungal therapy, with the main classes being azoles, echinocandins, and polyenes. Each antifungal class targets specific fungal cellular components, but they face challenges like growing drug resistance and patient toxicity [6,7,8,9,10]. Hence, there is a pressing need for new antifungals.

In a quest to address this challenge, various strategies have been exploited to develop new drugs. High-throughput screening (HTS) and rational drug design are two ways to identify leads for drug development [11,12]. For HTS, target-based and whole-cell-based screenings are two major approaches, and both require investigators to screen thousands of compounds, which requires enormous physical resources, both chemically and mechanically [13]. On the contrary, rational drug design develops drugs based on information from the structure, function, and mechanism of action of the target protein and can also aid in optimizing hits identified from HTS [11]. Rational drug design also comes in two types: ligand-based and structure-based approaches. The former depends on understanding the structure of existing ligands that can bind to a target, while the latter focuses on designing inhibitors using the structural details of target proteins [14]. The rational drug design process typically involves multiple rounds of design, synthesis, and evaluation to yield compounds potent and specific enough for preclinical trials [15]. The atomic structures, as well as predicted structures from various methods, of the target proteins have been shown to be useful in drug design [16,17]. Dorzolamide, used for treating glaucoma [15], and saquinavir, an HIV protease inhibitor [18], are two market drugs refined or conceived through the structure-based method.

In recent years, minimal progress has been made to identify new antifungals because of the relatively high gene homology (~40%) and conservation of fundamental biochemical pathways between fungi and humans. This similarity prohibits easy identification of drugs that are selectively toxic to fungi [19]. To this end, it is important to identify fungal-specific targets that are either absent in mammals, or can be inhibited specifically enough not to cause side effects in mammals, and can thus provide selective inhibition of fungi. The most effective of these molecules in current use all target the cell envelope (cell wall–cell membrane complex). These include the antifungal classes of echinocandins, azoles, allylamines, and polyenes, which act on fungal cell walls, cell membrane biosynthesis, and membrane integrity, respectively. There are other inhibitors known to target non-envelope proteins as well, but they are less commonly used in treating invasive infections in humans. They target nucleic acid biosynthesis, the respiratory chain, and microtubule function. For example, pyrimidine analogs, such as flucytosine, inhibit fungal thymidylate synthase, affecting DNA and RNA synthesis [20]. However, the major drawbacks of flucytosine include widespread occurrence of resistance in many fungal species and bone marrow toxicity in patients [21,22,23,24]. Benzimidazoles, such as thiabendazole, disrupt microtubule function, leading to an inhibition of fungal cell division and, ultimately, cell death [25]. Thiabendazole has been used to treat a variety of plant fungal infections, and to a lesser extent, to treat fungal infection in animals and humans because of its narrow spectrum of activity and the potential for liver toxicity [26,27]. Similarly, succinate dehydrogenase inhibitors (SDHIs) and quinone outside inhibitors, targeting succinate dehydrogenase and cytochrome bc1 complex (complex III), respectively, disrupt energy production in fungi and have been used in agriculture to combat molds and fungi [28,29]. In addition, novel agents like olorofim (targeting dihydroorotate dehydrogenase) and fosmanogepix (targeting inositol acyltransferase) possess broad spectrum activity and remarkable novelty that are expected to be significant in the future [30,31,32,33,34].

Among all current and potential antifungal drug targets, membrane-bound enzymes involved in cell wall and cell membrane biosynthesis have been the most valuable targets based on success and use for invasive infections. Thus, these organelles likely still represent the best targets for the development of novel antifungal agents due to their essential nature in fungal survival, proliferation, pathogenicity, and resistance to antifungal drugs. In addition, some of these enzymes are absent in humans. Here, we provide an overview of the structural biology of several membrane-bound enzymes considered current or potential antifungal targets, as well as known inhibitors, and their potential use in rational drug design. These targets all affect the cell envelope of fungi, which comprises the cell wall and cell membrane.

## 2. Cell Wall Biosynthesis Enzymes

The fungal cell wall is an ideal target for antifungal drugs as it is an organelle that is not conserved in mammals. The cell wall shields fungi from environmental threats and prevents harmful macromolecules from entering the cell [35]. The fungal cell wall accounts for around 40% of the entire cell volume and is made of polysaccharides (mainly glucan and chitin) and glycoproteins [36]. Structurally, chitin and β-1,3-glucan are essential constituents of most fungal cell walls, and they create a gel-like matrix that in some fungi can include α-1,3-glucans, β-1,6-glucans, and glycoproteins. The synthesis of chitin and glucan is mediated by membrane-bound chitin synthases and glucan synthases, respectively, which are effective targets for antifungal drugs. Table 1 lists the antifungal drugs or inhibitors on the market or in the developmental stage that target membrane-bound enzymes involved in cell wall biosynthesis.

### 2.1. Chitin Synthases

Chitin accounts for 1–2% of the dry weight of the yeast cell wall and could reach up to 10–20% in filamentous fungi [62]. In *C. albicans*, chitin content in the hyphae wall is three times higher than that of the yeast form [63]. Chitin is a long-chain polymer consisting of β-(1,4)-linked N-acetylglucosamine (GlcNAc), and because of its absence in plants and vertebrates, the biosynthesis of chitin is considered a promising target for antifungal drugs [64]. The chitin layer is formed by large families of plasma membrane-bound chitin synthases, which catalyze the formation of β(1–4) bonds using UDP-GlcNAc as the sugar source [65,66]. There are a total of seven classes of chitin synthases, and class IV enzymes often generate the majority of the cell wall chitin in fungi and are generally associated with virulence [66,67,68]. *S. cerevisiae* has three chitin synthase genes (*CHS1*, *CHS2*, and *CHS3*) and *C. albicans* has four (*CHS1*, *CHS2*, *CHS3*, and *CHS8*), while *Aspergillus nidulans*, *Aspergillus fumigatus*, and *Cryptococcus neoformans* are known to have eight [66].

Polyoxin B (PolyB) is a type of peptidyl nucleoside that acts against chitin synthases as a competitive inhibitor [37]. It has been employed for many years in the fields of agriculture and forestry to combat fungal plant pathogens and harmful arthropods (which have chitinous exoskeletons) [37,38]. Similarly, nikkomycin Z (NikZ) is another peptidyl nucleoside that inhibits chitin synthase, and it has demonstrated substantial positive effects in treating fungal infections in mammals [40,41]. In 2000, a range of new inhibitors for *Ca*Chs1 was discovered through an extensive screening process, which led to the discovery of the compound RO-09-3024, a very effective chitin synthase inhibitor with an IC_50_ value of 0.14 nM in vitro and an EC_50_ of 70 μg/mL against *C. albicans* (CY1002) [69]. However, many fungal pathogens contain *CHS* genes that are less sensitive to these inhibitors, stressing the need to optimize these molecules via further drug design, which requires chitin synthase structures [39,70]. However, as a multi-transmembrane enzyme, chitin synthases have proven challenging for protein expression, solubilization, and crystallization, hindering structural analysis [71]. For this reason, a bacterial glycosyltransferase from *Sinorhizobium meliloti*, *Sm*NodC, is shown to be an appropriate model to study the general structure and reaction mechanism of chitin synthases due to the fact that (i) *Sm*NodC has a catalytic core that is conserved with chitin synthases [72,73] and (ii) *Sm*NodC is inhibited by nikkomycin Z [71]. The homology models of *Sm*NodC and *Sc*Chs2 were made based on the structure of bacterial cellulose synthase from *Rhodobacter sphaeroidesi* [74], and have generally similar structural architectures. One difference between them is that *Sm*NodC is missing the chitin transport channel present in *Sc*Chs2 [71]. A detailed display of the active site and product-binding site of *Sm*NodC is shown in [71].

The first atomic structure of chitin synthase was solved from the soybean root rot pathogenic oomycete *Phytophthora sojae* in 2022 via cryo-EM [42]. The structure of this chitin synthase was solved in apo-, GlcNAc-bound, UDP-bound (post-synthesis), nascent chitin oligomer-bound, and most importantly, nikkomycin Z-bound forms (Figure 1). *Ps*Chs shares great sequence and architectural similarity with *Sc*Chs, but with an elevated *K*_i_ value for nikkomycin Z [42]. This could represent the binding mode of nikkomycin Z to fungal chitin synthase. As a substrate analog, nikkomycin Z binds to the uridine-binding tub via its uridine segment in the same way as the substrate UDP-GlcNAc does (Figure 1B). The hydroxypyridine moiety of nikkomycin Z occupies a significant portion of the reaction chamber and translocating channel. This restricts the donor substrate from accessing the reaction area required for chitin synthesis. The hydroxypyridine ring also forms hydrophobic interactions with Leu412, Tyr433, Val452, Pro454, and Trp539 from the conserved motifs in *Ps*Chs. The mutation of these residues impairs activity but, in the meantime, reduces inhibition from nikkomycin Z [42].

Predicted structures from models are powerful, but actual solved structures are more informative. A structure of *C. albicans* chitin synthase 2 (class I) was recently solved [39]. Structures were solved for the apo-, substrate-bound, nikkomycin Z-bound, and polyoxin D-bound forms of *Ca*Chs2. Similarly, nikkomycin Z and polyoxin D occupy the substrate binding site of *Ca*Chs2, and an overlay of bound UDP-GlcNAc and polyoxin D with nikkomycin Z is shown in Figure 2. For nikkomycin Z, the aminohexuronic acid moiety (red arrow) occupies an overall similar position as UDP-GlcNAc (Figure 2A). However, nikkomycin Z gains interactions with residues Y571 and W647 on the pyridinyl ring, which are absent from UDP-GlcNAc. However, nikkomycin Z lacks or has severely decreased interaction with residue D465, as this residue rotates away when bound to nikkomycin Z. In contrast, polyoxin D adapts a slightly different binding mode compared to nikkomycin Z. The critical residue involved in polyoxin D binding is Q643, forming two hydrogen bonds with hydroxyl groups on polyoxin (Figure 2A). Residue K440 also rotates to interact with polyoxin D, which is not seen in either nikkomycin- or UDP-GlcNAc-bound forms. This interaction between K440 and the 5-carboxyl of the uracil base is suggested to be the additional inhibition mechanism that polyoxin has on *Ca*Chs2 activity [39]. It was suggested that the stronger inhibitory effect of nikkomycin Z on *Ca*Chs2 compared to polyoxin D probably results from the enhanced interaction via the pyridinyl ring. The presence of the 5-carboxyl in polyoxin D and its interaction with K440 somewhat compensates for the absence of interactions associated with the pyridinyl ring [39].

The third and most current structure of a fungal chitin synthase is *S. cerevisiae* Chs1, (class I) [75]. Again, the structures of substrate-, polyoxin D- and nikkomycin Z-bound *Sc*Chs1 were determined using cryo-EM, and the mechanism of chitin synthesis initiation, extension, and transport was described in [75]. One unique finding on the mode of polyoxin D and nikkomycin Z binding is that besides the competition from the nucleoside moiety on the UDP, the peptidyl moiety of polyoxin D and nikkomycin Z opens the switch loop and thus keeps the gate of chitin transport channel blocked [75]. This unique mechanism of inhibition can potentially be used in future rational drug design.

### 2.2. β-1,3-Glucan Synthase

β-glucan is the predominant polysaccharide in fungal cells, constituting ~50–60% of their dry weight [62]. Moreover, 65–90% of these glucan polymers have a ß-1,3 linkage, but there exist other linkage types, like β-1,6 (in *Candida* spp.), β-1,4, α-1,3, and α-1,4. Among these different linked glucans, the most structurally significant component is β-1,3-D-glucan, which serves as the anchor for other covalent attachments within the wall [62,76]. β-1,3-D-glucan is synthesized by glucan synthases, a group of membrane-bound enzymes located in the plasma membrane. Glucan synthases have a conserved catalytic domain (Fks) and are regulated by Rho1 GTPase subunits [77,78]. The genes *FKS1* and *FKS2*, responsible for producing β-1,3-D-glucan synthases, were first discovered in *Saccharomyces cerevisiae* [78,79], and later the orthologs were identified in other fungal species. Disruption of one *FKS* gene leads to cell growth perturbation and disruption of both causes cell death in *S. cerevisiae* [35,79,80], indicating that they were a promising drug target.

Echinocandins, derivatives of secondary metabolites from *Aspergillus nidulans* and *Aspergillus rugulosus*, act as non-competitive inhibitors of β−(1-3)-glucan synthase [45]. The primary cellular mode of action of echinocandins is associated with disruption of the fungal cell wall, which is then vulnerable to osmotic imbalances, leading to the death of the fungal cell and a reduction in damage to the host tissue [46,47]. The mode of action of echinocandins in the host may also be associated with the host immune response, specifically increased detection of ß-(1,3)-glucan by the pathogen receptor dectin-1 [81]. The FDA-approved echinocandin-class drugs are caspofungin, micafungin, anidulafungin, and rezafungin (Figure 3), with rezafungin being approved by the US Food and Drug Agency recently in 2023 [9,82,83]. Structurally, they are all lipopeptides with similar cores, and one noticeable difference among them is the side chain. The long fatty acid chain was hypothesized to disrupt the membrane and thus inhibit glucan synthase activity [9,82], and the resistance mutations of echinocandins often occur at the highly conserved hot spot 1 (HS1, residue 641–649, TM5), hot spot 2 (HS2, residue 1345–1365, TM8), and hot spot 3 (HS3, 690–700, TM6) [84,85,86]. In 2023, the structures of *S. cerevisiae* FKS1 and the echinocandin-resistant mutant, S643P FKS1, were determined using cryo-EM [48], providing structural insights into the mechanism of echinocandin resistance. In *Sc*FKS1, the active site is located in the interface between the cytoplasm and plasma membrane, with a putative path for glucan translocation spanning across the membrane layers [48]. HS 1, 2, and 3 are shown in Figure 4A and are spatially located very close to each other. In wildtype *Sc*FKS1, the residues F639 and S643 from the HS1 region play roles in lipid binding, as the side chain of F639 has direct interactions with three lipid molecules, while the side chain of S643 seems to stabilize the lipid-binding residue Y638 (Figure 4B). However, in the echinocandin-resistant S643P mutant, the side chains of both F639 and Y638 rotate significantly, leading to the re-orientation of bound lipids (Figure 4C). Therefore, the echinocandin resistance mechanism was hypothesized to be that the re-oriented amino acid side chains and corresponding lipid movement lead to a change in [i] the binding site of echinocandins or [ii] *Sc*FKS1’s response to membrane perturbation caused by echinocandin [48]. Later, Zhao et al. also reported the structure of *Sc*FKS1, and proposed an echinocandin resistance mechanism [49]. In this report, the interface formed by TM5, TM6, and TM8 undergoes conformational changes during glucan transport with TM8 shifting outward. However, it was suggested that instead of re-orientating lipid molecules, the echinocandin-resistant mutants S643P and S643Y might escape inhibition either by improving catalytic efficiency or disrupting drug binding [49]. This minor discrepancy can potentially be explained by different sample preparation procedures (e.g., different detergents used), but the structures from these two studies are very similar overall. Evidence of more direct drug/protein interactions will help determine the mechanism of echinocandin inhibition and resistance, aiding in designing more potent echinocandin-class drugs.

## 3. Cell Membrane Biosynthesis Enzymes

Drugs that impact cell membrane integrity have seen significant success as well [87]. For example, drugs targeting ergosterol synthesis, such as imidazoles and triazoles, are especially effective. While many new antifungal triazole compounds have been introduced recently, they still have a long journey before being recognized as successful antifungals [88]. Beyond sterols, the membrane also contains essential components like phospholipids and sphingolipids, which are crucial for cellular operations and signaling pathways. Here, we will summarize current and potential membrane-bound enzyme antifungal targets involved in ergosterol, sphingolipid, and phospholipid biosynthesis, with their available structures and inhibitors. Table 2 lists antifungal drugs or inhibitors on the market or in developmental stages that target membrane-bound enzymes involved in cell membrane biosynthesis.

### 3.1. Ergosterol Biosynthesis Enzymes

#### 3.1.1. Lanosterol 14α-Demethylase (Erg11)

Ergosterol is a major component of fungal cell membranes and plays a role similar to cholesterol in human cell membranes [89,90,91]. Its biosynthesis pathway in fungi is a prime target for antifungal drug development, and a detailed ergosterol biosynthesis pathway including all involved enzymes and the sites of inhibition are summarized in [89]. These enzymes, residing in the endoplasmic reticulum and other organelles, are crucial for the synthesis of ergosterol, the main sterol in fungal cell membranes. For example, lanosterol 14α-demethylase (known as Erg11 or CYP51) is a single-pass membrane-bound cytochrome P450 enzyme and is a well-studied drug target of azoles [92,93]. Erg11 is responsible for the demethylation of lanosterol, a step vital for the subsequent conversion of lanosterol to ergosterol, inhibition of which leads to the accumulation of toxic intermediates, thus compromising cell membrane integrity [93]. The structures and their interactions with different azoles were identified using X-ray crystallography for *Saccharomyces cerevisiae*, *Aspergillus fumigatus*, *Candida albicans*, and *Candida glabrata* Erg11 [92,94,95,96]. Like other members within the P450 enzyme family, Erg11 has a thiolate–heme iron center, the active site where oxidation reactions occur (Figure 5B). Also, it contains hydrophobic pockets and channels that accommodate the lipid substrate and facilitate its access to the catalytic center [92,94,95,96]. The interaction between itraconazole and *C. albicans* Erg11 shows that on a molecular scale, one of the nitrogen atoms in the azole ring binds as the sixth coordinating ligand to the heme iron, preventing oxygen activation (Figure 5B) [97]. The interaction of the azole ring with the heme is crucial in determining the binding of azole drugs to Erg11 targets in other structures as well [94,95,96]. With the detailed structural information of Erg11 available, structure-directed drug discovery can be performed, which involves virtually screening compound libraries for molecules that might bind more effectively to Erg11, even in resistant enzymes, or designing new molecules based on insights from the enzyme’s structure [97].

**Table 2 jof-10-00171-t002:** Antifungal drugs or inhibitors targeting membrane-bound enzymes in cell membrane biosynthesis.

Drug Class/Agent	Structure of an Exemplar Compound	Target Enzyme	Mechanism of Action	Discovery Stage	Is the Atomic Structure Solved for the Target?	Is the Drug–Target Interaction Known?	Reference
**Azoles (e.g., fluconazole, itraconazole)**	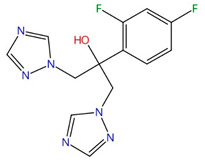 Fluconazole	Lanosterol 14α-demethylase (Erg11)	Inhibit ergosterol biosynthesis	Approved	Yes	Yes	[92,94,95,96]
**Allylamines (e.g., terbinafine)**	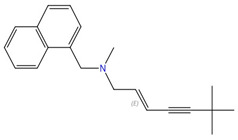 Terbinafine	Squalene epoxidase (Erg1)	Inhibit ergosterol biosynthesis	Approved for treating topical and oral fungal infections	Yes	Yes	[98,99,100,101]
**Tomatidine**	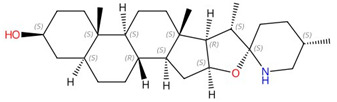	C-24 sterol methyltransferase (Erg6)	Inhibits ergosterol biosynthesis	Research and development	No	No	[102]
**Arylguanidines (e.g., abafungin)**	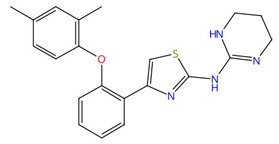 Abafungin	C-24 sterol methyltransferase (Erg6)	Inhibit ergosterol biosynthesis	Research and development	No	No	[103]
**H55**	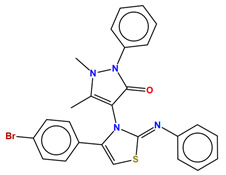	C-24 sterol methyltransferase (Erg6)	Inhibits ergosterol biosynthesis	Research and development	No	No	[104]
**Morpholines (fenpropimorph, fenpropidin, amorolfine, and Sila-analogue 24)**	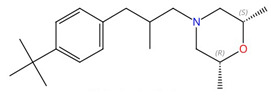 Fenpropimorph	Sterol C-14 reductase (Erg24) and sterol C-8,7 isomerase (Erg2)	Inhibit ergosterol biosynthesis	Research and development	No	No	[105]
**Sphingofungins**	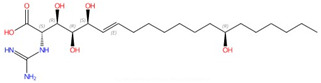 Sphingofungin A	Serine palmitoyltransferase (SPT)	Inhibit sphingolipid biosynthesis	Research and development	No	No	[106,107]
**Lipoxamycin**	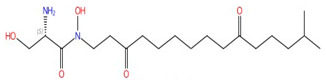	Serine palmitoyltransferase (SPT)	Inhibits sphingolipid biosynthesis	Research and development	No	No	[108,109]
**Fumonisins (e.g., fumonisin B1)**	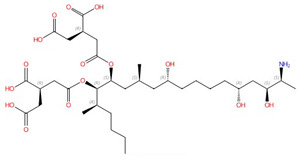 Fumonisin B1	Ceramide synthase	Inhibit sphingolipid biosynthesis	Research and development	No	No, but a model was proposed in [110]	[110,111,112]
**Rustmicin**	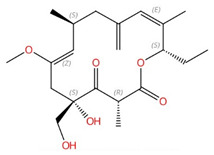	Inositol phosphorylceramide (IPC) synthase	Inhibits sphingolipid biosynthesis	Research and development	No	No	[113,114]
**Khafrefungin**	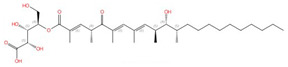	Inositol phosphorylceramide (IPC) synthase	Inhibits sphingolipid biosynthesis	Research and development	No	No	[115]
**Aureobasidin A**	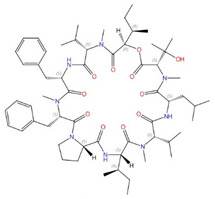	Inositol phosphorylceramide (IPC) synthase	Inhibits sphingolipid biosynthesis	Research and development	No	No	[116,117]
**Haplofungin**	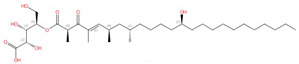	Inositol phosphorylceramide (IPC) synthase	Inhibits sphingolipid biosynthesis	Research and development	No	No	[118,119]
**YU253467 and YU254403**	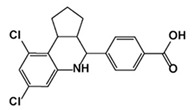 YU254403	Phosphatidylserine decarboxylase	Inhibit phospholipid biosynthesis	Research and development	No	No	[120]
**CBR-5884**	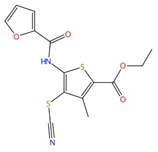	Phosphatidylserine synthase	Inhibits phospholipid biosynthesis	Research and development	No	No	[121]

#### 3.1.2. Squalene Synthase (Erg9)

Besides Erg11, other membrane-bound enzymes are also involved in ergosterol biosynthesis. For example, squalene synthase (Erg9), an enzyme that catalyzes the conversion of two molecules of farnesyl pyrophosphate to squalene, is also a potential drug target due to the functional difference between animal and fungal counterparts [122,123,124]. The structures of Erg9 homologs from a trypanosomatid parasite, *Trypanosoma cruzi*, and humans were first determined via X-ray crystallography; this could aid in the development of anti-Chagas disease and cholesterol-lowering drugs [125,126,127]. Zaragozic acids are potent competitive inhibitors of rat liver squalene synthases and can potentially treat hypercholesterolemia [128]. Malwal et al. reported the first fungal Erg9 structure from *Aspergillus flavus* in both apo- and substrate-bound forms and compared it to previous structures [122]. The transmembrane domains of *A. flavus* Erg9 have similar architectures compared to their human counterparts, but the B-helix is significantly shorter in human Erg9 [122]. This difference might lead to different ligand/inhibitor binding between human and pathogen proteins and could be used in antifungal design.

#### 3.1.3. Squalene Epoxidase (Erg1)

Following the formation of squalene, squalene epoxidase (also known as Erg1) adds an epoxide group to squalene to form 2,3-oxidosqualene [89,122]. Erg1 is also a membrane-bound enzyme located in the ER membrane and its reaction is the rate-limiting step of ergosterol biosynthesis in fungi and cholesterol biosynthesis in mammals. It is predicted to form a complex with Erg9 in the microsomal fraction [129,130]. Terbinafine, an allylamine drug (Figure 6A, Table 2), inhibits Erg1 and leads to ergosterol depletion and accumulation of squalene, which is fungicidal for filamentous fungi but fungistatic for most *Candida* species [98,99,100]. The human Erg1 structure was solved with an N-terminally truncated enzyme (118–574) in the presence of a known inhibitor NB-598 [101]. The terbinafine molecule was superposed with NB-598 to show the potential mode of inhibition of the molecule (Figure 6B). The conserved residues of human Erg1, L326, L473, F477, F492, F495, L508, P505, and H522 are predicted to form non-polar interactions with the inhibitor, and the mutations of these equivalent residues lead to terbinafine resistance in fungi [101,131,132,133]. Furthermore, a homology model of *S. cerevisiae* Erg1 was made and compared to its human counterpart [129]. *S. cerevisiae* Erg1 possesses an extended loop between β-strands 6 and 7 (residues 109–139, based on *S. cerevisiae*, pointed by the arrow), while the human homolog has a much shorter loop (residues 210–220) (Figure 6C). This compact loop in the human version might aid in crystal formation and the extended loop in *S. cerevisiae* Erg1 might obstruct this process [129]. Currently, the function of the extended loop in *S. cerevisiae* Erg1 is unknown, but may be targeted to develop molecules that destabilize the protein or disrupt potential protein–protein interactions.

#### 3.1.4. C-24 Sterol Methyltransferase (Erg6)

C-24 sterol methyltransferase (known as Erg6) is also a membrane enzyme involved in ergosterol biosynthesis which was suggested to be an antifungal target due to its absence in mammals [89,134]. Erg6 catalyzes the methylation of the 24th carbon in the sterol side chain in the later stages of ergosterol biosynthesis, the disruption of which leads to reduced mating capability, diminished tryptophan uptake, increased permeability, and susceptibility to cations and antifungals in *S. cerevisiae* [135,136,137]. In *C. albicans*, the disruption of Erg6 leads to increased sensitivity to cycloheximide, terbinafine, fenpropimorph, and tridemorph, but not to azoles, while showing resistance to amphotericin B [138]. Interestingly, deletion of Erg6 leads to reduced virulence but not cell growth [139].

Several sterol analogs were determined to suppress Erg6 activity due to their structural resemblance to the substrate or product of Erg6 [140]. Other inhibitors of Erg6, such as tomatidine and arylguanidines (Table 2), effectively hinder the growth of *C. albicans*, but might inhibit additional cellular targets since disruption of Erg6 does not lead to growth defects [102,103,139,141]. Recently, an antipyrine derivative, H55 (Table 2), identified from screening, showed low cytotoxicity and effectively inhibited *C. albicans* hyphal formation under various conditions, and also exhibited therapeutic efficacy in mouse models of azole-resistant candidiasis [104]. Various assays support the hypothesis that H55 is an allosteric inhibitor for Erg6, and a molecular dynamics simulation predicts that H55 competes with S-adenosylmethionine for binding to Erg6 [104]. More structural information is needed to validate or provide more insight into the interaction between Erg6 and H55.

There are currently no structures experimentally solved for Erg6 to our knowledge, but Azam et al. modeled a C-24 sterol methyltransferase from *Leishmania infantum* and identified relevant residues that interact with itraconazole and amphotericin B [134]. Since the substrate-binding sites and active sites are conserved between *L. infantum* and *S. cerevisiae* C-24 sterol methyltransferase, ligand/protein interaction information from the *L. infantum* Erg6 homolog can potentially be applied in antifungal design [134].

#### 3.1.5. Sterol C-14 Reductase (Erg24) and Sterol C-8,7 Isomerase (Erg2)

Morpholines (Table 2) are known to inhibit sterol C-14 reductase (known as Erg24) and sterol C-8,7 isomerase (Erg2), which are two membrane-bound enzymes involved in ergosterol biosynthesis [89,142,143]. Morpholines such as fenpropimorph, fenpropidin, and amorolfine, as well as a silicon containing analog named Sila-analogue 24, exhibit potent antifungal effects against different human fungal pathogens [105]. An *erg24∆∆* mutant has reduced virulence in a mouse model of disseminated candidiasis [144]. Erg24 catalyzes the reduction of the C14=15 double bond of sterol intermediates, so sterol intermediates that are not processed by Erg24 cannot be recognized by downstream enzymes, thus perturbing the membrane [89]. Erg2 facilitates the formation of a double bond from the 8 to the 7 position in the sterol intermediate fecosterol, and this enzyme has a polyvalent high-affinity drug binding site similar to that in mammalian sigma receptors [145]. Currently, neither the structures of Erg24 and Erg2 nor their interactions with morpholines have been characterized. The human Erg2 homolog has a solved structure, and its interaction with the anti-breast cancer drug tamoxifen and the cholesterol biosynthesis inhibitor U18666A have been studied [146].

Another sterol C-14 reductase is Erg23, and one bacterial homolog, *Methylomicrobium alcaliphilum* sterol C-14 reductase, show an interesting arrangement of ten transmembrane domains, with the catalytic domain localized in the carboxy-terminal half (TM6–10). This domain surrounds two linked pockets, with one facing the cytoplasm, which accommodates NADPH, and the second pocket accessible from the lipid bilayer [147]. However, neither the structure of human sterol C-14 reductase nor *Methylomicrobium alcaliphilum* sterol C-14 reductase have direct use in the antifungal design, and efforts are needed in structure determination of their fungal counterparts.

### 3.2. Sphingolipid Biosynthesis Enzymes

Sphingolipid production is crucial for the growth and survival of various human fungal pathogens, such as *Histoplasma capsulatum* and *C. albicans* [148,149]. Therefore, using a drug to obstruct this process could effectively halt their growth and trigger cell death. The sphingolipid biosynthesis pathway *in S. cerevisiae* is depicted in Figure 7 from the first step to the formation of the major sphingolipid mannose-(inositol-P)_2_-ceramide (M(IP)_2_C) [117,150,151]. Three membrane-bound enzymes involved in sphingolipid synthesis have been suggested as potential antifungal targets—serine palmitoyltransferase (SPT), ceramide synthase, and inositol phosphorylceramide (IPC) synthase. These all have their respective inhibitors.

#### 3.2.1. Serine Palmitoyltransferase (SPT)

SPT catalyzes the condensation of serine and palmitoyl-CoA, which is the first and rate-limiting enzyme in the biosynthesis of sphingolipids [152,153]. SPT uses pyridoxal phosphate (PLP) as a cofactor for catalysis, and belongs to the allene oxide synthase (AOS) family [154]. The active yeast SPT enzyme is a heterodimer made from subunits encoded by *lcb1* or *lcb2* [155,156], and a third subunit, Tsc3p, is required for high-level SPT activity [157]. The protein structure of the SPT complex was first solved in bacteria [158,159,160], but since the bacterial SPT homologs are soluble homodimers, they provided limited insights into the catalytic mechanism of eukaryotic SPT. In 2021, structures of the human SPT–ORMDL3 complex (ORMDL proteins function as regulatory subunits) in different catalytic states were solved [161,162], and later, the ceramide-sensing mechanism of the SPT-ORMDL3 complex was studied from a ceramide-bound structure [163]. Due to the high sequence similarity between fungal and mammalian SPT subunits [164], the human SPT-ORMDL3 complex can be used to study the mechanism of SPT inhibitors such as sphingofungins [106,107] and lipoxamycin [108,109] (Table 2).

#### 3.2.2. Ceramide Synthase

Ceramide synthase, another membrane-bound enzyme involved in sphingolipid biosynthesis, adds a fatty acyl chain from fatty acyl–coenzyme A (CoA) to the sphingoid base sphinganine to form ceramide [117]. Mammals possess six ceramide synthase isoforms that differ in their tissue distribution and substrate specificity, and each isoform is known to produce ceramides with different acyl chain lengths [165,166]. Currently, there is no structure available for any fungal ceramide synthase homologs to our knowledge, but some studies have provided structure-function characterization of ceramide synthases. Ceramide synthase belongs to the longevity assurance gene 1 (Lag1) protein family, which has a stretch of 52 amino acids that form a highly conserved Lag1p motif [167]. Two conserved histidine residues within this Lag1p motif are crucial for the catalysis and binding of the substrates, the alteration of which negatively impacts the enzymatic function of mammalian and yeast ceramide synthases [167,168,169]. Fumonisins are a group of mycotoxins that have a striking structural resemblance to sphingolipids and are carcinogenic [111] (Table 2). Notably, fumonisin B1 (FB_1_), among many fumonisins, effectively inhibits ceramide synthase, conferring toxicity and carcinogenic properties. Although neither the protein structure of ceramide synthase nor the FB_1_/protein binding is solved, an inhibition model for FB_1_ was proposed [110]. Briefly, concentrations of both substrates affected the potency of FB_1_, suggesting that FB_1_ is a competitive inhibitor that binds to the active site of ceramide synthase [112]. It was later found that the tricarballylic acid sidechains play essential roles in the inhibition of FB_1_, as eliminating tricarballylic acid sidechains reduces the strength of ceramide synthase inhibition in vitro by 10-fold [110,170]. Moreover, this model was further supported by the observation that the FB_1_ derivative without tricarballylic acid sidechains can be used as a substrate by ceramide synthase, indicating those side chains are required for inhibition [171]. Further structure investigation is needed to validate this model and will also help optimize FB_1_ to act specifically against fungi.

#### 3.2.3. Inositol Phosphorylceramide (IPC) Synthase

Unlike serine palmitoyltransferase and ceramide synthase, which have homologs in mammalian cells, inositol phosphorylceramide (IPC) synthase catalyzes a reaction unique to plants and some microbial eukaryotes, such as fungi and kinetoplastids. This reaction is the transfer of phosphoinositol from phosphatidylinositol to phytoceramide [117]. Following its discovery in *S. cerevisiae*, IPC synthases have been characterized in plants [172,173,174] and various protozoans causing neglected tropical diseases, such as Chagas disease and leishmaniasis [175,176,177,178,179]. An alignment of trypanosomatid IPC synthases showed conserved arginine, histidine, and aspartate residues in the active site, and their contributions to a predicted catalytic transfer of the phosphoryl group were demonstrated in *Leishmania major* IPC synthase [179].

Four inhibitors that act specifically against IPC synthases have been identified (Table 2). Rustmicin, a 14-membered macrolide, is especially potent against *C. neoformans*, where it inhibits the growth of *C. neoformans* and its sphingolipid synthesis at concentrations <1 ng/mL, with an IC_50_ of 70 pM against solubilized *C. neoformans* IPC synthase [113,114]. The compound khafrefungin, isolated from a Costa Rican plant, displayed antifungal effects again *C. albicans* and *C. neoformans* and was determined to inhibit *C. albicans* IPC synthase with an IC_50_ of 0.6 nM, but with no effects on mammalian sphingolipids [115]. Another compound, aureobasidin A, a natural compound from the fungus *Aureobasidium pullulans*, has very low (sub-μg/mL) MIC values for *S. cerevisiae*, *C. albicans*, and *C. neoformans* with IC_50_ values for IPC synthase activities ranging from 0.2 to 4.9 nM [116,117]. Haplofungin, a phytoceramide mimic isolated from the fungus *Lauriomyces bellulus*, also showed potent inhibitory activities against fungal IPC synthases [118,119]. However, despite the fact that several potent IPC synthase inhibitors have been identified, the atomic structure of this protein is unsolved. The *Arabidopsis thaliana* IPC synthase monomer is predicted to have six transmembrane domains with a flexible N-terminal region (AlphaFoldDB: Q9SH93), and further structure–activity relationship studies will be helpful for optimizing current inhibitors or designing new antifungal drugs.

### 3.3. Phospholipid Biosynthesis Enzymes

Phospholipids, accounting for 40–60% of lipids in eukaryotic cells, are the predominant lipids present in most organisms’ membranes [180]. The four major phospholipids in eukaryotes are phosphatidylserine (PS), phosphatidylcholine (PC), phosphatidylethanolamine (PE), and phosphatidylinositol (PI) [181]. PC and PE constitute the majority of phospholipids in yeasts and are required for functional membrane construction in eukaryotic organisms. They are involved in membrane integrity and mitochondrial functions [182,183]; PI species are involved in various cellular signal transduction pathways [184]. PS is enriched in the plasma membrane [181] and is involved in a variety of other signaling cascades such as the activation of protein kinase C [185,186,187]. PS is required for virulence in *C. albicans* and viability in *C. neformans* [183,188,189], but also plays important roles in apoptosis and blood clotting in mammals [190,191,192]

Phospholipid biosynthesis in cells is intricate, with distinct variations between fungal and mammalian cells. The phospholipid biosynthesis pathways of *C. albicans* (A) and mammals/parasites (B) are shown in Figure 8, adapted from [193]. The biosynthesis of phospholipids in *C. albicans* differs from mammalian cells in several steps. First, mammalian cells encode one *PSD* gene for PS decarboxylase [194], which converts PS to PE, while yeast has two distinct proteins with little similarity, *PSD1* and *PSD2*. Each of these genes has PS decarboxylase activity [195,196]. Also, the production of PS uses a different mechanism in mammalian versus fungal cells. In mammalian cells, PS is produced through a base-exchange reaction catalyzed by the mammalian phosphatidylserine synthase-1 (PSS1) and phosphatidylserine synthase-2 (PSS2), in which the headgroups of existing PC and PE, respectively, are cleaved off and replaced with serine to produce PS [197]. On the contrary, fungal cells condense cytidine diphosphate diacylglycerol (CDP-DAG) and serine into PS via phosphatidylserine synthase (PS synthase), using a different catalytic mechanism compared to the mammalian PSS1/PSS2 enzymes [181,197]. Here, we will discuss the current inhibitors and structure studies of PS decarboxylase and PS synthase.

#### 3.3.1. PS Decarboxylases (PSD)

In both yeast and mammals, PE is synthesized through the de novo pathway by decarboxylating PS or through the Kennedy pathway by using exogenous ethanolamine (Figure 8). The Kennedy pathway contributes to the majority of PE in some mammalian cells [198,199,200], but in yeast, the majority of PE is generated by the decarboxylation of PS [194,196,201]. Also, research has shown that while the elimination of the Kennedy pathway does not impact yeast cell survival, disruption of the *PSD1* gene leads to ethanolamine auxotrophy and mitochondrial instability [196]. PSDs are evolutionarily conserved across a wide range of organisms, and most are membrane-associated enzymes relying on a covalently attached pyruvoyl moiety for their activities [202]. Membrane-bound PSDs are synthesized as a single polypeptide proenzyme, which undergoes self-cleavage at a highly conserved LGST motif [203,204]. The α and β chains from the cleavage assemble into a mature PSD homodimer, with each protomer having one α and one β chain. The α chains function as the catalytic domains and β chains facilitate membrane association [202]. Recently, the structures of apo and PE-bound *E. coli* PSDs were solved, and structural insight into detailed mechanisms of membrane-binding, PS recognition, self-cleavage, and catalysis were provided [205,206].

Several PSD inhibitors have been proposed and are shown in Table 3. In the 1970s, hydroxylamine was found to inhibit the enzymatic activity of PSD and PE synthesis, and induce the accumulation of PS [207,208]. The effects of hydroxylamine are similar when incubated with *S. cerevisiae* and *C. albicans*, leading to an accumulation of PS and decreases in PE and PC, but PC and PE levels are much lower in *C. albicans* compared with *S. cerevisiae* [209]. Similarly, serine hydroxamate, a serine analog, was also found to inhibit the conversion of PS to PE in *E. coli* with an accumulation in PS, indicating that it targets PS decarboxylase [210]. However, neither the specificity nor inhibition mechanism have been described for hydroxylamine and serine hydroxamate. In 2007, a screen of a collection of 9920 molecules was performed against human inner mitochondrial membranes containing the PSD enzyme, where direct measurements of PS and PE were generated. This screen identified 54 molecules that exhibited inhibition in a dose-dependent manner [211]. More recently, one molecule was identified from a cell-based screening and it is 7-chloro-N-(4-ethoxyphenyl)-4-quinolinamin (MMV007285), which has potent inhibition of *Plasmodium falciparum* PSD, with low toxicity toward mammalian cells [212]. An analog of this compound, 7CPQA, also exhibited inhibition of *Pf*PSD activity [212]. Later, two compounds, YU253467and YU254403, were discovered from a target-based screen, and they inhibit both native *C. albicans* growth and PSD mitochondrial activity [120]. The molecules identified from these different screens are promising, but the addition of detailed protein structure and ligand interactions would improve optimization efforts for higher specificity and potency.

#### 3.3.2. PS Synthase

Fungi use the Cho1 PS synthase to catalyze the formation of PS from CDP-DAG and serine, and both the enzyme and the reaction are absent in mammals [181,197,217], indicating a potential antifungal target. Furthermore, deleting the PS synthase in *C. albicans* prevents it from causing disease in mouse models of oral or systemic candidiasis [183,218]. PS synthase is also crucial for the growth of the major fungal pathogen *C. neoformans* [189] and is also highly conserved across various fungal species [217]; these observations indicate that PS synthase is an excellent drug target.

PS synthases were first characterized in bacteria. The PS synthases from Gram-negative bacteria such as *E. coli*, *Salmonella typhimurium*, and *Enterobacter aerogenes* are tightly associated with ribosomes, and perform catalysis when they bind to the plasma membrane [219]. Gram-positive bacteria such as *B. megaterium*, *Bacillus subtilis*, and *Clostridium perfringens* have membrane-associated PS synthase, which have conserved motifs and belong to the protein family that includes eukaryotic counterparts [220]. The first eukaryotic PS synthase was identified in *S. cerevisiae* [221,222]. Since then, the characterization of *S. cerevisiae* PS synthase (Cho1) included understanding the regulation of Cho1 [223,224,225,226,227], identifying the localization of the enzyme [228,229], protein solubilization and purification [230,231], and determination of Michaelis–Menton kinetics [223,230,231]. The *C. albicans* PS synthase was first characterized in 2010, with the finding that it is crucial for systemic *Candida* infections in mice [183]. Michaelis–Menton kinetics of *C. albicans* PS synthase were described, and its conserved CAPT motif for binding CDP-DAG was identified as well as some residues involved in serine binding [232,233,234]. Later, *C. albicans* PS synthase was solubilized and purified, but surprisingly formed a hexamer. It is unique among the known structures of the same family of membrane-bound phospholipid synthases, which are all dimers [235].

PS synthase belongs to the CDP-alcohol phosphatidyltransferase (CDP-AP) protein family, and six prokaryotic [236,237,238,239,240,241] and two eukaryotic [242,243] members have solved structures to date. Among these, there is only one PS synthase structure, and it is from the archaean *Methanocaldococcus jannaschii*, and has eight transmembrane domains [241]. This is different from the homology model of *C. albicans* PS synthase with six transmembrane domains. In addition, some key residues involved in catalytic activity of *C. albicans* PS were lacking functions in *M. jannaschii* PS synthase [235,241]. A structure of fungal PS synthase will reconcile this discrepancy and also provide insights into the mechanisms of *C. albicans* PS synthase catalysis.

Currently, despite the fact that PS synthase is a promising drug target, the identification of its specific inhibitors is in the early stage. Two cell-based screens were conducted to identify inhibitors of *C. albicans* PS synthase, but both attempts were unsuccessful [244,245]. One screen pinpointed the compound SB-224289. However, it was later determined that SB-224289 only acts on PS synthase-related physiological pathways rather than the enzyme directly [244]. The other screen identified bleomycin, but it was also found that this compound affects phospholipid-associated processes rather than targeting *C. albicans* PS synthase directly [245]. Recently, Zhou et al. identified a molecule, CBR-5884 (Table 3), that inhibits both purified PS synthase and its function in vivo, with a *K*_i_ of 1550 ± 245.6 nM [121]. This molecule acts as a competitive inhibitor for serine, thus having the potential for further development. However, more efforts are needed to identify additional inhibitors to this promising drug target that can potentially lead to new classes of antifungals.

#### 3.3.3. Other Miscellaneous Phospholipid Synthesis Inhibitors

Finally, besides PS decarboxylase and PS synthase, several inhibitors have been identified to target phospholipid biosynthesis pathways for PI and PC. For example, it was suggested that validamycin A might hinder the incorporation of inositol into PI in *Rhizoctonia cerealis*, a process driven by the membrane-associated enzyme PI synthase [213], but a detailed enzymatic characterization is missing. For de novo PC biosynthesis, where PE is methylated three time into PC (Figure 8), ethionine and 2-hydroxyethyl-hydrazine were shown to inhibit PE methylation, and thus resulted in lower PC levels [214,215]. Moreover, the anticancer compounds BR23 and BR25, known to inhibit human choline kinase, directly inhibited the ethanolamine activity of *P. falciparum* choline kinase, thus significantly reducing PE levels in *P. falciparum* without affecting PC [216]. This led to halted growth of the parasite due to the depletion of membrane PE, and was ultimately lethal [216]. These observations underscore the significance of phospholipid biosynthesis in certain microbes’ survival and pathogenicity and thus drug development.

## 4. Conclusions

The role of structural biology has expanded significantly in antifungal drug discovery, employing advanced techniques like X-ray crystallography, NMR spectroscopy, and cryo-electron microscopy (cryo-EM) to reveal intricate details of enzyme–inhibitor interactions. These methods have revolutionized our understanding by providing high-resolution images of target enzymes and their detailed interactions with inhibitors, facilitating the design of novel inhibitors with improved specificity and potency. This detailed molecular insight has been key to overcoming drug resistance and toxicity challenges, paving the way for more effective antifungal therapies. Besides the advanced techniques for solving protein structures, bioinformatics tools, such as homology modeling and newly developed AlphaFold, may also predict the structures of target enzymes when the experimental structures are not available. These techniques could deepen our knowledge of structure–activity relationships, refining antifungal drug design strategies and speeding up the discovery of new antifungals.

The unique properties of the fungal cell wall and membrane make these organelles ideal targets for antifungal agents, as they are essential to the survival of fungi and also contain enzymes not found in human cells, making it possible to discover molecules targeting fungal-specific pathways without affecting human cells [8,82]. However, a drawback to targeting cell wall and cell membrane biosynthesis enzymes is the potential for the rapid development of resistance by fungi, necessitating continuous research into novel targets [8,82]. Future directions include leveraging advanced genomics and proteomics to identify unique fungal enzymes and pathways involved in cell wall and membrane biosynthesis and employing structure-based drug design to develop more effective and selective antifungal agents. Additionally, a more profound understanding of protein–ligand interactions between current antifungals and novel drugs with their respective targets could also lead to the optimization of current drugs and the development of new drugs, addressing resistance issues and improving treatment outcomes. In sum, this review not only sheds light on the structural intricacies of membrane-bound antifungal targets and their inhibitors but also suggests targets and pathways for future explorations in structural biology, which are crucial for the advancement of rational drug design and the development of more effective antifungal therapies.

## Figures and Tables

**Figure 1 jof-10-00171-f001:**
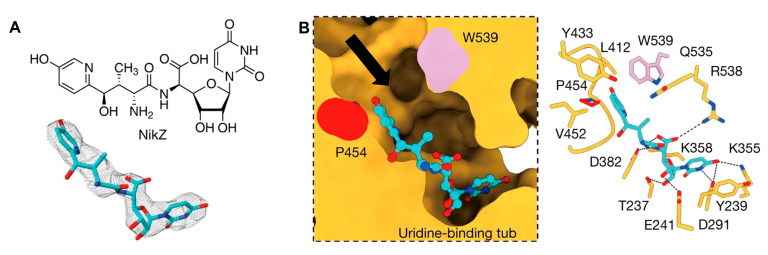
The interaction between nikkomycin Z and Phytophthora sojae Chs1. (**A**) Chemical and 3D structure of nikkomycin Z (NikZ). (**B**) The left is a sliced-surface view of the NikZ-binding site of PsChs and the right is detailed interactions between NikZ and PsChs1. The reaction chamber and translocating channel are pointed out by the arrow. Hydrogen bonds are labeled as black dashed lines. Figures originally generated in [42] (under Creative Commons CC BY license) and adapted for this review.

**Figure 2 jof-10-00171-f002:**
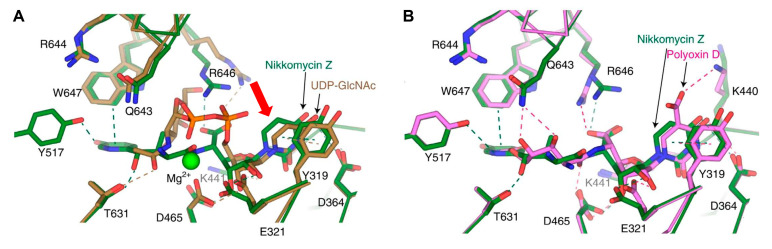
Binding models of UDP-GlcNAc, nikkomycin Z, and polyoxin D to *C. albicans* Chs2. (**A**) Overlay of substrate binding sites: one with UDP-GlcNAc (in brown) and the other with nikkomycin Z-bound (in green) in *Ca*Chs2. The aminohexuronic acid moiety is noted by a red arrow. (**B**) Overlay of the substrate binding sites of *Ca*Chs2: one bound with nikkomycin Z (in green) and the other with polyoxin D (in magenta). Hydrogen bonds and π-π stacking interactions between the substrate or ligand and *Ca*Chs2 are marked with dashed lines in their respective colors. Figures originally generated in [39] and adapted for this review with permission (License Number: 5697420844040).

**Figure 3 jof-10-00171-f003:**
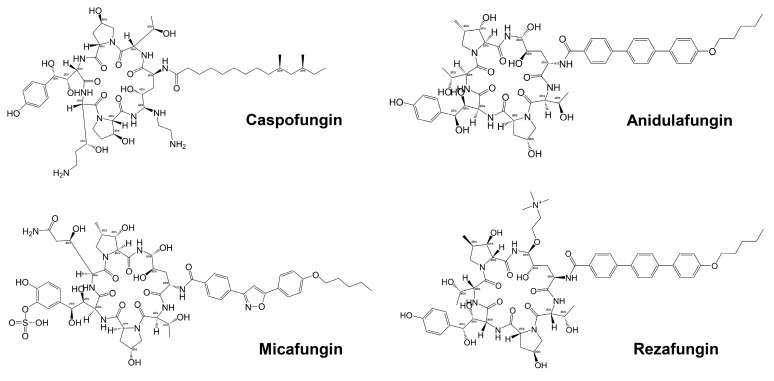
Chemical structures of the three FDA-approved echinocandins.

**Figure 4 jof-10-00171-f004:**
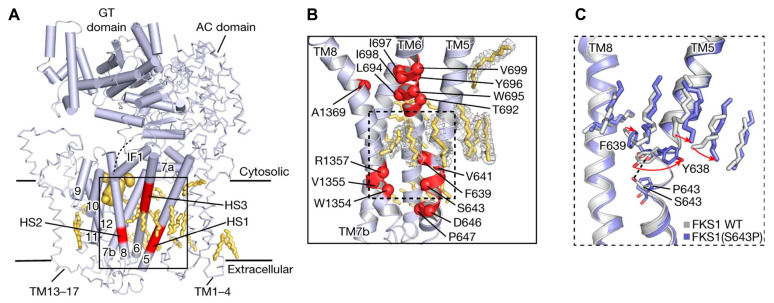
Structural interpretation of echinocandin-resistant mutations in ScFKS1 structure. (**A**) The ScFKS1 structure with three distinct hotspot regions (colored in red) labeled as HS1–3. These regions are associated with mutations that confer resistance to echinocandins. (**B**) A detailed view of echinocandin-resistant mutations is provided, as referenced in (**A**). The mutations’ alpha carbon (Cα) atoms are illustrated as red spheres. (**C**) Conformational changes and lipid re-arrangements, marked by red arrows, in wildtype ScFKS1 (grey) and drug-resistant mutation S643P ScFKS1 (blue). Potential polar interaction is indicated by the black dashed line. Figures were originally generated in [48], and are reused in this review with permission (License Number: 5697430533684).

**Figure 5 jof-10-00171-f005:**
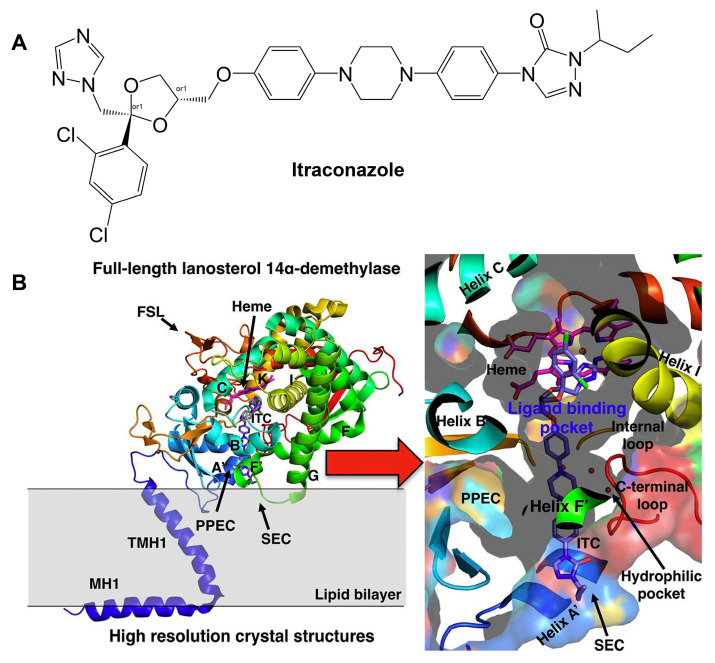
Binding of itraconazole to S. cerevisiae Erg11. (**A**) Structure of itraconazole. (**B**) S. cerevisiae Erg11 structure originally printed in [97] and reused in this review with permission (License Number: 5697681298825). The left-hand image shows a cartoon representation of the overall fold of S. cerevisiae Erg11 and its predicted position in the lipid membrane(PDBID:5EQB). The right-hand image shows the binding of itraconazole within the S. cerevisiae Erg11. Itraconazole is shown in purple and heme moiety is shown in pink. (ITC: itraconazole; FSL: fungus-specific loop; SEC: substrate entry channel; PPEC: putative product exit channel; LBP: ligand-binding pocket; MH1: amphipathic helix; TMH1: transmembrane helix).

**Figure 6 jof-10-00171-f006:**
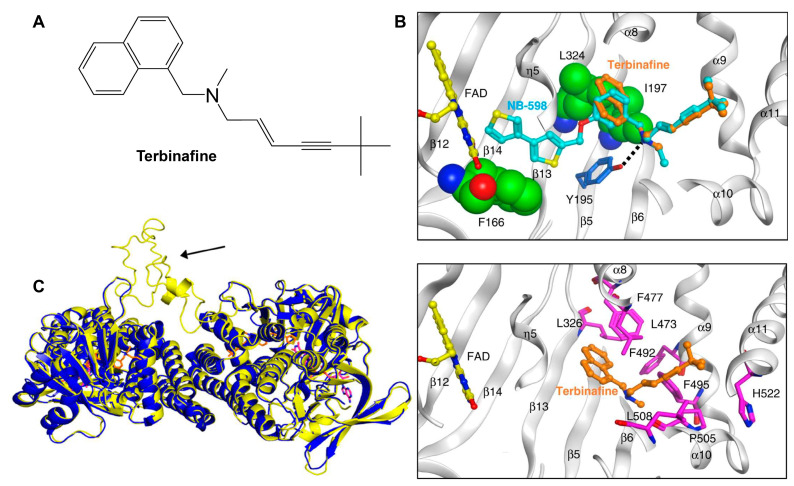
Binding model of terbinafine to squalene epoxidase. (**A**) Structure of terbinafine. (**B**) The upper panel shows a superposition of terbinafine (orange) with NB-598 (cyan) in human squalene epoxidase (PDBID:6C6P). The lower panel shows the positions of the known terbinafine-resistant mutations (pink) with respect to terbinafine (orange) in human squalene epoxidase with a superposed terbinafine model. (**C**) Superposition of human squalene epoxidase structure (blue) and S. cerevisiae squalene epoxidase homology model (yellow). The extended loop on S. cerevisiae Erg1 is pointed to by an arrow. Figure (**B**) was originally generated in [101] (under Creative Commons CC BY license), and Figure (**C**) was originally generated in [129] (under Creative Commons CC BY license).

**Figure 7 jof-10-00171-f007:**
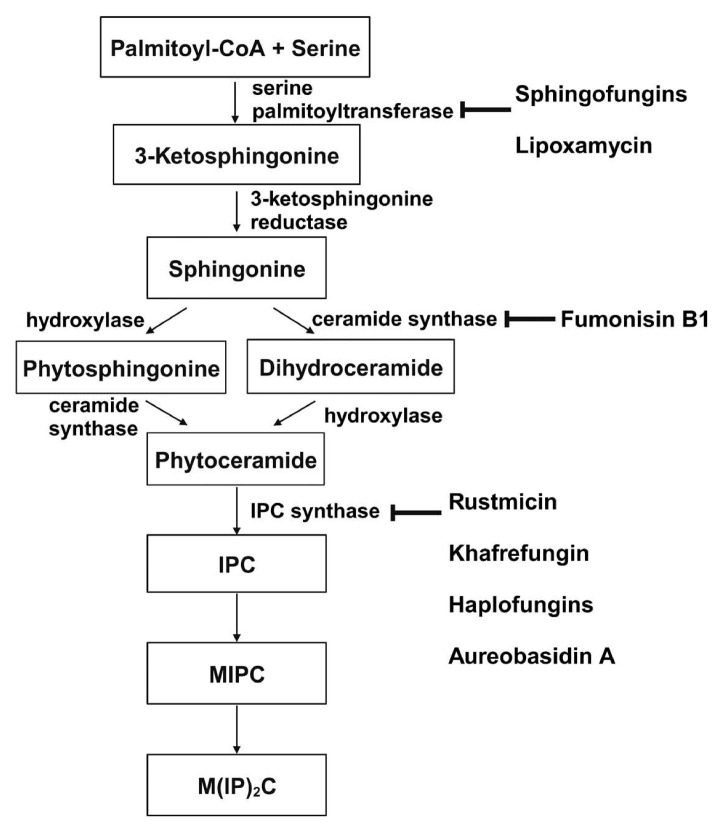
De novo synthesis of sphingolipids in S. cerevisiae and known inhibitors targeting each enzyme. This figure is adapted and modified from [91] (under Creative Commons CC BY license). IPC: Inositolphosphoryl-ceramide; MIPC: mannose inositol-P-ceramide; M(IP)_2_C: mannose-(inositol-P)_2_-ceramide.

**Figure 8 jof-10-00171-f008:**
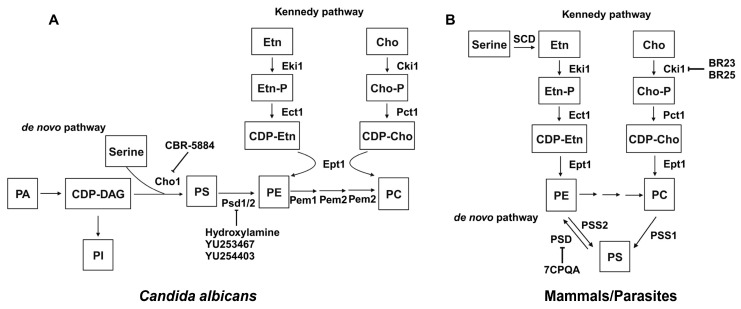
Phospholipid biosynthesis pathways for C. albicans (**A**) and mammals/parasites (**B**). Known inhibitors to certain targets are also shown. Both the de novo pathway and Kennedy pathway exist in each scenario. This figure is adapted and modified from [193] (under Creative Commons CC BY license). PA: phosphatidic acid; CDP-DAG: cytidine diphosphate diacylglycerol; Ser: serine; Cho1/PSS1/PSS2: PS synthase; PI: phosphatidylinositol; PS: phosphatidylserine; PE: phosphatidylethanolamine; PC: phosphatidylcholine; Etn: ethanolamine; Cho: choline; Etn-P: phosphoethanolamine; Cho-P: phosphocholine; CDP-Etn: cytidyldiphosphate-ethanolamine; CDP-Cho: cytidyldiphosphatecholine; PSD: PS decarboxylase; Eki1: ethanolamine kinase; Ect1: ethanolamine-phosphate cytidylyltransferase; Ept1: ethanolamine phosphotransferase; Cki1: choline kinase; Pct1: choline-phosphate cytidylyltransferase; SDC: serine decarboxylase; Cpt1: choline phosphotransferase.

**Table 1 jof-10-00171-t001:** Antifungal drugs or inhibitors targeting membrane-bound enzymes in cell wall biosynthesis.

Drug Class/Agent	Structure of an Exemplar Compound	Target Enzyme	Mechanism of Action	Discovery Stage	Is the Drug–Target Interaction Known?	Reference
**Polyoxins**	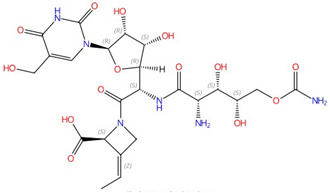 Polyoxin A	Chitin synthase	Inhibit chitin synthesis in cell wall	Research and development	Yes	[37,38,39]
**Nikkomycins**	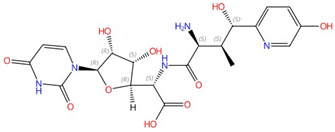 Nikkomycin Z	Chitin synthase	Inhibit chitin synthesis in cell wall	Clinical trials	Yes	[39,40,41,42]
**Arthrichitin**	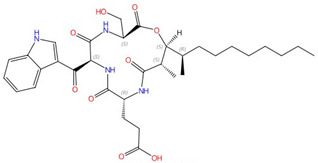	Chitin synthase	Inhibits chitin synthesis in cell wall	Research and development	No	[43]
**Radicicol**	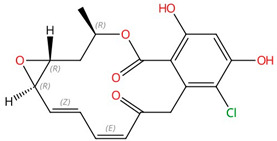	Chitin synthase	Inhibits chitin synthesis in cell wall	Research and development	No	[44]
**Echinocandins (caspofungin, micafungin, anidulafungin and rezafungin)**	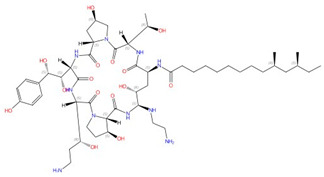 Caspofungin	β-1,3-glucan synthase	Inhibit cell wall glucan synthesis	Approved	Yes	[45,46,47,48,49]
**Ibrexafungerp**	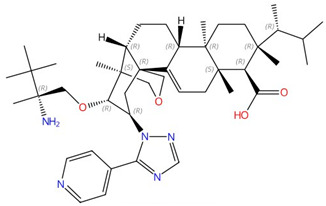	β-1,3-glucan synthase	Inhibits cell wall glucan synthesis	Approved for treating vulvovaginal candidiasis	No	[50,51]
**Pneumocandin A-E**	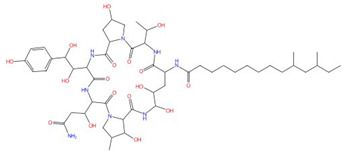 Pneumocandin A_0_	β-1,3-glucan synthase	Inhibits cell wall glucan synthesis	Research and development	No	[52,53,54]
**Aculeacin A-G**	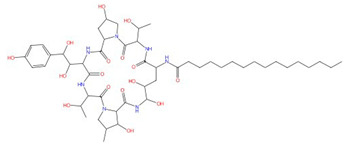 Aculeacin A	β-1,3-glucan synthase	Inhibits cell wall glucan synthesis	Research and development	No	[55,56]
**Mulundocandin**	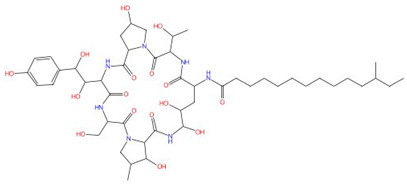	β-1,3-glucan synthase	Inhibits cell wall glucan synthesis	Research and development	No	[57,58]
**Enfumafungin**	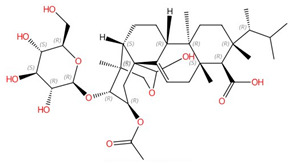	β-1,3-glucan synthase	Inhibits cell wall glucan synthesis	Research and development	No	[59]
**Arundifungin**	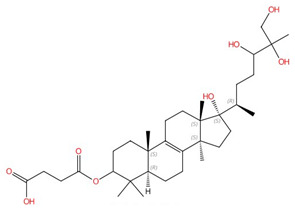	β-1,3-glucan synthase	Inhibits cell wall glucan synthesis	Research and development	No	[60]
**Papulacandins**	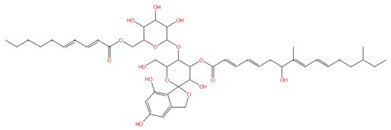	β-1,3-glucan synthase	Inhibit cell wall glucan synthesis	Research and development	No	[61]

**Table 3 jof-10-00171-t003:** Potential phospholipid biosynthesis enzymes as drug targets and inhibitors.

Compound	Target Enzyme	Organism	Reference
Hydroxylamine	PS decarboxylase	*E. coli*	[207,208]
Hydroxylamine	PS decarboxylase	*S. cerevisiae* and *C. albicans*	[209]
Serine hydroxamate	PS decarboxylase	*E. coli*	[210]
7CPQA	PS decarboxylase	*P. falciparum*	[212]
YU253467 and YU254403	PS decarboxylase	*C. albicans*	[120]
CBR-5884	PS synthase	*C. albicans*	[121]
Validamycin A	PI synthesis	*R. cerealis*	[213]
Ethionine	PE methylation	*S. cerevisiae*	[214]
2-hydroxyethyl-hydrazine	PE methylation	*S. cerevisiae*	[215]
BR23 and BR25	Choline kinase	*P. falciparum*	[216]

## Data Availability

This review synthesizes data from previously published studies and publicly available sources, with appropriate citations and permissions provided. It also includes modified figures that have been enhanced to offer a more comprehensive visual representation of the data discussed. These enhancements are based on existing data and aim to support the review’s analytical depth without generating new data. For non-publicly accessible sources, readers are advised to refer to the original publications or contact the original authors. Further inquiries can be directed to the corresponding author.

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
