# Peer review of "Innovations in Antifungal Drug Discovery among Cell Envelope Synthesis Enzymes through Structural Insights"

_jof, 2024, doi:10.3390/jof10030171_

Round 1

Reviewer 1 Report

Comments and Suggestions for Authors

The review „Adding structure to the search for new antifungal drugs among cell envelope synthesis enzymes” covers the issues of describing new inhibitors of enzymes important for the synthesis of fungal cell wall components in order to use them in novel potential therapeutic approaches. The manuscript is detailed but concise, well written, and there are only some minor comments.

In the first fragment in the Introduction section, in lines 24-28, the phrase “fungal infections in humans” is repeated three times. This section should be rearranged to avoid this repetition.

Table 1 has distorted formatting, perhaps at the submission stage, please pay attention to it because it is difficult to follow in the first column. The similarly constructed tables would be very helpful for readers also in other subsections concerning other described groups of enzymes.

Please indicate for each figure if there is permission from the authors or the journal to reuse the scheme and please specify the license.

Author Response

The review „Adding structure to the search for new antifungal drugs among cell envelope synthesis enzymes” covers the issues of describing new inhibitors of enzymes important for the synthesis of fungal cell wall components in order to use them in novel potential therapeutic approaches. The manuscript is detailed but concise, well written, and there are only some minor comments.

In the first fragment in the Introduction section, in lines 24-28, the phrase “fungal infections in humans” is repeated three times. This section should be rearranged to avoid this repetition.

Thanks for pointing this out. It has been addressed.

Table 1 has distorted formatting, perhaps at the submission stage, please pay attention to it because it is difficult to follow in the first column. The similarly constructed tables would be very helpful for readers also in other subsections concerning other described groups of enzymes.

The line numbering has been fixed for the table. And two more tables are added  for section 1 and 2.

Please indicate for each figure if there is permission from the authors or the journal to reuse the scheme and please specify the license.

This information has been added to the figure legends.

Reviewer 2 Report

The authors of the manuscript titled "Adding structure to the search for new antifungal drugs among cell envelope synthesis enzymes" summarize the structures of several current and potential membrane-bound antifungal targets involved in cell wall and cell membrane biosynthesis and their interactions with known inhibitors or drugs. The body of work presented here is appropriate for the Journal of Fungi; however, it needs some major revisions before it can be considered for publication.

Points that need to be addressed.

1. In the introduction section, the authors should add a paragraph regarding all the known antifungal drug discovery targets. Also, highlight the pros and cons of different known targets for antifungal drugs.

2. For both section 2 and section 3, the authors should provide us with a table containing the enzymes along with drugs that are already in the market targeting them along with compounds that are in various stages of drug discovery (clinical, preclinical, and research and development). Also, provide information regarding whether the target enzyme structures are already known. Add the structures of the compounds in the development stages to the table.

3. Before the conclusion section, the authors should add a section for structural biology highlighting various methods used to study target enzymes and their interactions with novel inhibitors and drugs. Also please highlight how novel methods influenced the antifungal drug discovery

4. In the conclusion section, in addition to summarizing the whole topic, the authors need to provide their insight into why targeting cell wall and membrane-bound proteins is a better strategy compared to other fungal drug targets. Also, highlight the pros and cons of targeting cell envelope synthesis enzymes. Please provide a future direction where this field of research would lead.

None.

Author Response

The authors of the manuscript titled "Adding structure to the search for new antifungal drugs among cell envelope synthesis enzymes" summarize the structures of several current and potential membrane-bound antifungal targets involved in cell wall and cell membrane biosynthesis and their interactions with known inhibitors or drugs. The body of work presented here is appropriate for the Journal of Fungi; however, it needs some major revisions before it can be considered for publication.

Points that need to be addressed.

  1. In the introduction section, the authors should add a paragraph regarding all the known antifungal drug discovery targets. Also, highlight the pros and cons of different known targets for antifungal drugs.

The known antifungal drugs and their targets, as well as the pros and cons, are included in the introduction (lines 70-87).

  1. For both section 2 and section 3, the authors should provide us with a table containing the enzymes along with drugs that are already in the market targeting them along with compounds that are in various stages of drug discovery (clinical, preclinical, and research and development). Also, provide information regarding whether the target enzyme structures are already known. Add the structures of the compounds in the development stages to the table.

The two tables containing inhibitors and targets, compound structures and development stage are included for section 2 and 3, as Tables 1 and 2, respectively.

  1. Before the conclusion section, the authors should add a section for structural biology highlighting various methods used to study target enzymes and their interactions with novel inhibitors and drugs. Also please highlight how novel methods influenced the antifungal drug discovery

This information is added to the conclusion (lines 746-758).

  1. In the conclusion section, in addition to summarizing the whole topic, the authors need to provide their insight into why targeting cell wall and membrane-bound proteins is a better strategy compared to other fungal drug targets. Also, highlight the pros and cons of targeting cell envelope synthesis enzymes. Please provide a future direction where this field of research would lead.

This information is added to the conclusion (lines 759-771).

Reviewer 3 Report

Comments and Suggestions for Authors

I liked the theme chosen for this review. It is really important to elucidate the biological structure of various enzymes to understand potential interactions with possible inhibitors, and their potential use in rational drug design

The article is well written, well organized and contains a number of eloquent examples for the chosen topic.

However, I don't think the title chosen is the most appropriate for the article.

I would suggest a rephrasing. 

Author Response

I liked the theme chosen for this review. It is really important to elucidate the biological structure of various enzymes to understand potential interactions with possible inhibitors, and their potential use in rational drug design

The article is well written, well organized and contains a number of eloquent examples for the chosen topic.

However, I don't think the title chosen is the most appropriate for the article.

I would suggest a rephrasing.

Thank you. We have revised the title.

Round 2

Reviewer 2 Report

The authors have addressed all the comment s satisfactorily.

All the comments were addressed.